# Determination of Oxygen Heterocyclic Compounds in Foods Using Supercritical Fluid Chromatography–Tandem Mass Spectrometry

**DOI:** 10.3390/foods12183408

**Published:** 2023-09-13

**Authors:** Giovanna Cafeo, Antonella Satira, Marina Russo, Monica Mondello, Paola Dugo

**Affiliations:** 1Department of Chemical, Biological, Pharmaceutical and Environmental Sciences, University of Messina, 98168 Messina, Italy; giovanna.cafeo@studenti.unime.it (G.C.); paola.dugo@unime.it (P.D.); 2Chromaleont S.R.L., c/o Department of Chemical, Biological, Pharmaceutical and Environmental Sciences, University of Messina, 98168 Messina, Italy; antonella.satira@chromaleont.it (A.S.); monicamondello2010@gmail.com (M.M.)

**Keywords:** oxygen heterocyclic compounds, coumarin, furocoumarins, polymethoxyflavones, supercritical fluid chromatography, SFC-QqQ/MS

## Abstract

The aim of this research was to determine oxygen heterocyclic compounds in twenty-six *Citrus*- and cinnamon-flavoured foods using supercritical fluid chromatography in combination with triple-quadrupole mass spectrometry (SFC-QqQ-MS). According to the authors’ knowledge, this is the first report on the determination of these molecules in foods by means of the SFC-QqQ-MS technique. The analytical technique normally used for their determination in foods is liquid chromatography coupled with a photodiode array detector. However, supercritical fluid chromatography is proving to be a valid alternative approach to investigating coumarins, furocoumarins and polymethoxyflavones. According to the results presented herein, each sample analysed showed the presence of molecules of interest. Coumarin was found in all the cinnamon-flavoured samples analysed in a low concentration. The presence of oxygen heterocyclic compounds in all the *Citrus*-flavoured samples, according to the label, comfirmed that the foods selected for this research article were prepared with *Citrus* fruits. Among the samples analysed, mandarin juice was the richest in bioactive compounds, representing a good source of polymethoxyflavones in a diet.

## 1. Introduction

Supercritical fluid chromatography (SFC) represents a powerful analytical technique for the analysis of macro- and micro-constituents in natural products [1]. Moreover, the coupling of this chromatographic technique with a detection system such as mass spectrometry (MS) increases its sensitivity, allowing the identification of unknown molecules. Alkaloids, cannabinoids, anthraquinones, carotenoids, phenols, flavonoids, oxygen heterocyclic compounds, lipids, lignans, saponins, terpenes and vitamins are among the molecules primarily investigated in foods using the SFC-MS technique [1].

The determination of oxygen heterocyclic compounds (coumarins, furocoumarins and polymethoxyflavones) has often been a topic of interest for researchers [2,3]. Oxygen heterocyclic compounds naturally occur in plants and plant-based foods belonging principally to the Apiaceae and Rutaceae families. *Citrus* fruits (lemon, bergamot, sweet orange, grapefruit and mandarin), cinnamon, celery, parsley, parsnips and turnips are the main source of coumarins, furocoumarins and polymethoxyflavones in the diet [4,5]. Numerous beneficial effects on the human body are associated with the daily consumption of these compounds. For example, antioxidant, antiallergic, anti-inflammatory and antibacterial properties are attributed to these molecules, as long as dietary supplements based on polymethoxyflavones are consumed [6,7]. However, alongside these beneficial effects, negative effects on human health due to the intake of coumarin have also been highlighted [8,9]. For this reason, a limit for coumarin daily consumption was proposed by the European Food Safety Authority (EFSA). In an opinion released in 2004 from the EFSA, it was reported that a tolerable daily intake of coumarin is 0.1 mg coumarin/kg body weight [10]. From this opinion, in 2008, the European Parliament indicated the maximum level of coumarin permitted in several foods (desserts, bakery ware and cereals) [11]. No limits were proposed for the daily consumption of all the other oxygen heterocyclic compounds present in foodstuffs. *Citrus* fruits, celery, parsley, parsnips and turnips do not normally contain coumarin [2,3,4], so the only food that could represent a problem for human health is cinnamon, since it contains a quantity of coumarin of 300–7000 mg Kg^−1^ [5]. But, in any case, cinnamon is generally used as an ingredient in desserts, bakery goods and cereals. A high daily intake of coumarin itself is obviously difficult to achieve, although consumers may exceed the recommended intake through the consumption of cinnamon-flavoured foods. Thus, developing an analytical method with the aim of tracking the content of oxygen heterocyclic compounds is of fundamental importance for consumer health.

Several research articles on the characterization of oxygen heterocyclic compounds in the food, cosmetic and pharmaceutical fields have been published in the last thirty years [2,3]. The analytical technique mainly used for their determination in foods is liquid chromatography coupled with a photodiode array or mass spectrometer detector [12]. In fact, most of the scientific articles present in the literature used the liquid chromatography technique (HPLC) for the quantification of coumarin and oxygen heterocyclic compounds both in cinnamon-flavoured foods and in *Citrus* fruits [4,13,14,15,16,17,18,19,20,21]. Coumarin was generally quantified in cinnamon-flavoured bakery products, cereals, chocolate, milk and certainly in spices [13,16]. Furthermore, coumarins, furocoumarins and polymethoxyflavones were investigated in several *Citrus*-flavoured foods, and turnips, parsley, celery and carrots [4]. The oxygen heterocyclic compound profiles of *Citrus* juices and alcoholic beverages have already been reported [14,15,17,18,19]. Moreover, the coumarin, furocoumarin and polymethoxyflavones content in *Citrus*-flavoured beers [20] and extra-virgin olive oils [21] has also been elucidated. Each of these research articles used liquid chromatography as an analytical technique coupled with a photodiode and/or a mass spectrometer detector. Researchers have reported conducting an extraction procedure before the HPLC analysis, using a non-polar solvent [13,16,18,19,22] or the QuEChERS (Quick, Easy, Cheap, Effective, Rugged and Safe) approach [4].

All the methods previously described are time-consuming and use organic solvents both for the extraction procedure and for the analytical method. On the other hand, thanks to its faster analysis time and low solvent consumption, the SFC technique is a greener technique and an alternative to conventional HPLC [23]. Moreover, SFC is proving to be a valid alternative approach to investigating coumarins, furocoumarins and polymethoxyflavones [24,25,26,27,28,29,30]. SFC was already successfully adopted for the analysis of eight furocoumarins in four cosmetics after sample extraction [28]. In any case, an extraction procedure is mandatory before SFC analysis.

The aim of this research was to determine the concentration of coumarins, furocoumarins and polymethoxyflavones in foods using the SFC-QqQ/MS technique. The present analytical technique has already successfully been employed by our research group for the quantification of coumarins, furocoumarins and polymethoxyflavones in cold-pressed *Citrus* essential oils and fragrances [24,25]. To the best of the authors’ knowledge, this is the first attempt to quantify twenty-eight oxygen heterocyclic compounds in foods in less than eight minutes with a low consumption of methanol as a mobile phase.

## 2. Materials and Methods

### 2.1. Chemicals

A total of 28 analytical standards among coumarins, furocoumarins and polymethoxyflavones were used. Aurapten, bergamottin, citropten, cnidicin, coumarin, epoxybergamottin, isoimperatorin, isopimpinellin, oxypeucedanin, oxypeucedanin hydrate, nobiletin, herniarin, isomeranzin, merazin, merazin hydrate, 5-geranyloxy-7-methoxycoumarin, sinensetin, tangeretin, tetra-O- methylscutellarein and 8-methoxypsoralen standards were purchased from Merck Life Science (Merck KGaA, Darmstadt, Germany). Byakangelicol, psoralen, byakangelicin, cnidilin, 8-geranyloxypsoralen, bergapten, phellopterin and epoxyaurapten standards were purchased from Herboreal Ltd. (Edinburgh, UK).

SFC analyses were carried out using LC-MS-grade methanol (MeOH) (Merck Life Science) and 4.8-grade carbon dioxide (CO_2_) (Rivoira, Messina, Italy). HPLC-grade ethyl acetate and ethanol (Merck Life Science) were used for the coumarin, furocoumarin and polymethoxyflavone extraction procedure for all the samples.

### 2.2. Samples and Sample Preparation

This research was carried out on the twenty-six samples listed in Table 1. Eight *Citrus* juices (lemon, lime, bergamot, citron, sweet and bitter orange, mandarin and pink grapefruit) were obtained by squeezing fresh fruits in our laboratory. Fruits were collected at the same stage of ripeness from trees grown in Sicily (Italy) in May 2023. All fruits were washed, dried and stored at +4 °C. The other samples (five jams, an infusion, three alcoholic beverages, three soft drinks and six bakery products) were bought in a local market. The infusion was prepared as suggested by the producer. To one filter bag weighing two grams, 200 mL of boiling water was added and then left for five minutes. All the bakery products were first dried at room temperature for 48 h and then milled with a grinder in order to obtain a powder.

All the samples were subjected to liquid–liquid or solid–liquid extraction of coumarins, furocoumarins and polymethoxyflavones before the analysis. The extraction procedure was the same as that previously optimized in our laboratory [18,19]. Briefly, 10 mL of each liquid sample (infusions, juices, liquors and nonalcoholic beverages) were extracted with 10 mL of ethyl acetate by shaking the samples in a separatory funnel. The extraction procedure was repeated three times with the same amount of solvent. Then, the three ethyl acetate phases were gathered and dried in an evaporator (Envì EZ-2, Genevac, Ipswich, UK). On the other hand, a solid–liquid extraction was carried out on jams and bakery products. Specifically, 10 g of biscuit/snack powder or jam were weighed on a falcon scale, added to 10 mL of ethyl acetate and sonicated for 15 min. The supernatant was pooled while the residue was extracted two more times. The three aliquots of extract were collected and dried with an evaporator (Envì EZ-2, Genevac, UK). All the residues were dissolved in 10 mL of ethanol prior to SFC-QqQ/MS analysis. Mandarin juice extract was diluted to 1:100 (*v*/*v*) with ethanol, while lime, bergamot and pink grapefruit juice extracts were diluted to 1:5 (*v*/*v*). All extracts from alcoholic beverages, bergamot and green mandarin soft drinks, jams, lemon snacks and lemon and bergamot biscuits were diluted to 1:20 (*v*/*v*) with ethanol before SFC-QqQ/MS injection. Each sample was analysed in triplicate.

### 2.3. Instrumental and Analytical Conditions

The SFC-QqQ-MS analyses were performed with the same instrumental setup and method previously developed and validated [24]. The instrument was a Nexera-UC system coupled with an LCMS-8050 triple-quadrupole mass spectrometer (Shimadzu, Kyoto, Japan).

The SFC system was equipped with a DGU-20A_5R_ degasser, an LC-30ADSF CO_2_ pump, two LC-20ADXR dual-plunger parallel-flow pumps, an SIL-30AC autosampler, a CTO-20AC column oven, an SFC-30A backpressure regulator and a CBM-20A communication bus module. Separation of OHCs was carried out on an Ascentis Express dimethylpentafluorophenylpropyl (F5) column (150 × 2.1 mm, 2.7 µm) using CO_2_ and MeOH as mobile phases A and B, respectively. The chromatographic run was carried out at a flow rate of 1 mL min^−1^ under the following gradient conditions: from 0 to 10 min, increasing from 2% to 10% of B. The make-up pump solvent was MeOH, and the make-up flow rate was 1 mL min^−1^. The oven temperature and the back pressure regulator (BPR) were set at 40 °C and 120 bar, respectively. The injection volume was 2 µL.

The MS system was operated in multiple reaction monitoring (MRM) acquisition mode using an atmospheric pressure chemical ionization (APCI) interface set in positive ionization mode. The mass parameters were as follows: interface temperature, 350 °C; DL temperature, 250 °C; heat block temperature, 200 °C; nebulizing and drying gas flow (N_2_), 3 and 5 L min^−1^, event time, 0.024 s for each event; acquisition time, 10 min for all targets.

Data acquisition was performed with LabSolutions software ver. 5.80. An automatic method, based on set retention times and Q transitions, was used for peak integration. OHC quantification was based on the creation of calibration curves for all 28 standards in a distilled lemon essential oil as a blank matrix, according to the method validated by Arigò et al. [24]. Briefly, ten different concentration levels of each analytical standard, in the linearity range 0.001–3 mg L^−1^ (0.001, 0.003, 0.005, 0.01, 0.05, 0.1, 0.5, 1, 2 and 3 mg L^−1^), were analysed using SFC-QqQ-MS (five replicates for each level). Figures of merit (Limit of Detection (LoD), Limit of Quantification (LoQ), accuracy, intra- and inter-day repeatability) were calculated according to Eurachem guide [31]. Briefly, the distilled lemon essential oil was spiked with each oxygen heterocyclic compound at a concentration of 0.01 mgL^−1^ and injected five consecutive times under the same analytical conditions optimized for the samples [24]. The LoQs were between 0.0015 and 0.1536 mg L^−1^. Repeatability was calculated as the percentage coefficient of variation (CV%) in terms of peak areas, and all values were within a maximum shift of 8%. Accuracy values highlighted that the method ensured correct quantitative evaluation because they were between 80.0 and 118.6% [24].

## 3. Results

Thanks to the previously validated SFC-QqQ-MS method [24], it was possible to quantify twenty-eight oxygen heterocyclic components present in twenty-six food samples, as reported in Table 1. The identification of compounds was confirmed via comparison with standard materials. Samples were subjected to solvent extraction with ethyl acetate, as described in Section 2.2 of the Materials and Methods section. A solvent extraction step was mandatory before SFC-QqQ-MS for both solid and liquid samples. Normally, for the HPLC analysis of samples with a composition comprising more than 95% water, an extraction procedure is not necessary [16]. But for SFC analytical methods, the presence of water is not recommended. So, an extraction procedure for juices, infusions and alcoholic beverages was necessary. The environmentally friendly analytical approach allowed a fast separation of twenty-eight oxygen heterocyclic components in less than eight minutes, with a low consumption of methanol (10.5 mL per analysis). Table 2, Table 3 and Table 4 report the quali-quantitative profile of all the samples analysed. Thanks to the use of a triple-quadrupole mass spectrometer detector, it was possible to identify and quantify molecules present in trace levels, considering the LoQ values (0.004–0.01 mg L^−1^) [24]. In Figure 1, the SFC-QqQ-MS chromatograms of selected analysed samples are presented. The data reported in Table 2, Table 3 and Table 4 show the average concentration value of three consecutive analyses of the same sample ± the standard deviation.

Table 2 shows the quantitative results for coumarins, furocoumarins and polymethoxyflavones present in lab-made juice samples. From the data reported in Table 2, it is possible to see that all the samples investigated exhibited the presence of molecules of interest. The juices analysed showed the same qualitative profile as the corresponding cold-pressed essential oils [2]. Most of the oxygen heterocyclic compounds are non-polar, and so were present in the juices in a different ratio than that found in the cold-pressed essential oils. Among the juices investigated, mandarin juice was the richest in oxygen heterocyclic compounds (110.54 mg L^−1^), specifically polymethoxyflavones. On the other hand, lemon, citron and bitter and sweet orange juices exhibited a total concentration of coumarins, furocoumarins and polymethoxyflavones under 3 mg L^−1^. Bergamot juice was characterized by the presence of bergapten and bergamottin in a different ratio than that found in the cold-pressed essential oil (1:2 vs. 1:10, respectively) [2]. This is probably due to the low water solubility of bergamottin. Lime juice exhibited a total concentration of oxygen heterocyclic compounds of 27.92 mg L^−1^. Bergamottin, byakangelicin, oxypeucedanin hydrate, herniarin and 5-geranyloxy-7-methoxycoumarin are the molecules that principally characterized the lime juice. Bergamottin and 5-geranyloxy-7-methoxycoumarin were the main compounds found in cold-pressed lime essential oil [2]. On the other hand, byakangelicin, oxypeucedanin hydrate and herniarin were present in this juice sample because these are the most hydrophilic coumarins and psoralens in lime fruit. Pink grapefruit juice was characterized by the presence of two coumarins, meranzin and isomeranzin. Aurapten and epoxybergamottin are normally oxygen heterocyclic compounds characteristic of cold-pressed grapefruit essential oil [2]. As for the other juices, aurapten and epoxybergamottin are non-polar molecules and so were present at low concentration levels (0.26 and 0.84 mg L^−1^, respectively). The quali-quantitative data on the juices analysed are in accordance with previously published literature data [4,15,18,19,20].

From the quantitative data presented herein on the oxygen heterocyclic compounds contained in the hand-squeezed juice analysed, mandarin juice was the only one that could represent an optimum source of bioactive molecules in a diet. Considering that regular consumption of polymethoxyflavones leads to beneficial effects on human health [6,7] and that the EFSA recommends a daily consumption of 200 mg of bioactive molecules [32], the ingestion of mandarin juice can help the consumer achieve a good daily intake of bioactive molecules. According to the data reported in Table 2, 200 mL of mandarin juice contained about 22 mg of polymethoxyflavones.

Table 3 presents the results achieved for infusions and alcoholic and nonalcoholic beverages. All the samples showed the presence of molecules of interest. As reported in Table 1, each beverage label analysed reported the ingredients employed in the food manufacturing process. The oxygen heterocyclic compound concentration detected was in accordance with the ingredients listed in the label. Cinnamon-flavoured beverages showed the presence of coumarin (0.22 mg L^−1^ in infusions, 3.55 mg L^−1^ in liquor).

Considering the recommendations of the EFSA (0.1 mg Kg^−1^ body weight) [10] for daily coumarin consumption, the concentration found in our samples could be considered safe. In fact, three cups of the infusion analysed provides 0.132 mg of coumarin, and the consumption of less than 50 mL of cinnamon liquor provides a coumarin intake of 0.178 mg. From Table 3, is possible to see that the infusion also contained nobiletin and tetra-*O*-methylscutellarein, in accordance with the presence of sweet orange in the ingredient list [2,3].

The qualitative profile of the limoncello sample was the same as that of cold-pressed lemon essential oil [2], as can be seen in Figure 1B. Additionally, this alcoholic beverage was made with lemon peel, as reported in the label. The quantitative data are in accordance with previously published data [18,19], with 8-geranyloxypsoralen and oxypeucedanin hydrate presenting as the most abundant compounds. The same assumption can be made for bergamino and the bergamot soft drink regarding their qualitative profile. Both samples showed an oxygen heterocyclic compound profile close to that of cold-pressed bergamot essential oil [2]. The total concentration of molecules of interest was higher in bergamino than in the bergamot soft drink. This is probably due to the amount and type of flavouring ingredients used for beverage preparation. The soft drink, as reported in the label, had a bergamot juice concentration of 13%, while in bergamino, the use of bergamot peels is indicated. This explains the difference in the total concentration of oxygen heterocyclic compounds between the two foods (24.03 mg L^−1^ bergamino and 4.71 mg L^−1^ in the bergamot nonalcoholic beverage). The *Citrus* nonalcoholic beverage sample analysed showed a very low content of only citropten and oxypeucedanin hydrate. These data do not agree with the label description, which reported lemon infusion and bergamot essential oil as ingredients. Finally, in the green mandarin nonalcoholic beverages, only polymethoxyflavones were quantified. This result suggests that these molecules may come from the ingredients reported in the label (mandarin infusion and essential oil) [2].

Also, for the foods reported in Table 4, each jam and/or bakery product showed the presence of oxygen heterocyclic compounds. Regarding the ingredients reported in the label, each foodstuff showed a qualitative profile coherent with the coumarin, furocoumarin and polymethoxyflavone content in the *Citrus* fruit employed for the preparation. All the bakery products highlighted a total concentration of molecules of interest below 1.5 mg Kg^−1^, except for mandarin and bergamot biscuits. This low amount was probably due to the use of “lemon flavour” for lemon bakery products, as reported in the label. The wording “lemon flavour” does not allow us to trace the origin of the ingredient, and therefore does not allow us to draw conclusions. Regarding mandarin and bergamot biscuits, the compounds identified and quantified were those typically present in mandarin and bergamot peels, respectively [2,3].

A concentration of 1.16 mg Kg^−1^ of coumarin was found in orange cinnamon jam, as can be seen in Figure 1A. Also, in this case, the coumarin concentration found in our sample could be considered safe. In fact, a daily consumption of 40 g of this jam causes the ingestion of 0.05 mg of coumarin. Moreover, the use of orange peels indicates the presence of nobiletin and tetra-O-methylscutellarein. Compared with the other jams analysed, the lemon one was the richest in oxygen heterocyclic compounds, showing a profile typical of lemon peels. On the other hand, the orange jam had the lowest content, with the presence of only nobiletin (2.00 mg Kg^−1^) and tetra-O-methylscutellarein (0.76 mg Kg^−1^). The use of lemon, mandarin and orange as ingredients in the preparation of the *Citrus* ginger jam explained the presence of polymethoxyflavones typical of mandarin and orange and coumarins and psoralens typical of lemon [2,3].

## 4. Conclusions

This research proposed, for the first time, an SFC-QqQ-MS approach for the analysis of coumarins, furocoumarins and polymethoxyflavones in cinnamon and *Citrus*-flavoured foods. The environmentally friendly SFC-QqQ-MS analytical approach described in this work allowed a fast separation of twenty-eight OHCs in twenty-six foods. Characterization was achieved in less than 8 min, with a low consumption of methanol. Each sample was analysed after solvent extraction.

Coumarin was found in all the cinnamon-flavoured samples analysed in a low concentration. For this reason, consumption of these cinnamon-flavoured foods could be considered safe for human health according to the EFSA regulation. The presence of oxygen heterocyclic compounds in all the *Citrus*-flavoured samples, according to the label, confirmed that the foods selected for this research article were prepared with *Citrus* fruits. Among the samples analysed, mandarin juice was the richest in bioactive compounds, representing a good source of polymethoxyflavones in a diet.

## Figures and Tables

**Figure 1 foods-12-03408-f001:**
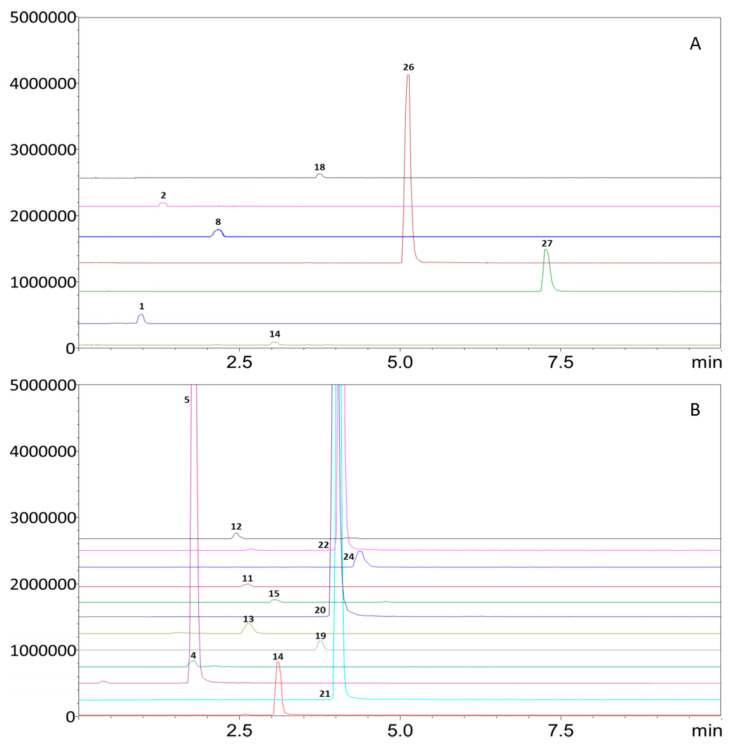
SFC-QqQ-MS (MRM acquisition mode) chromatograms of orange and cinnamon jam (**A**) and limoncello (**B**) samples. For peak identification, see Table 2 and Table 3.

**Table 1 foods-12-03408-t001:** List of the samples analysed.

Samples	Description (Label)
Juices	
Lemon (*Citrus limon* [L.] Burm.)	Lab-made *Citrus* juice
Lime (*Citrus aurantifolia* [Christm.] Swing.)	Lab-made *Citrus* juice
Bergamot (*Citrus bergamia,* Risso)	Lab-made *Citrus* juice
Citron (*Citrus medica,* L.)	Lab-made *Citrus* juice
Sweet orange (*Citrus sinensis* [L.] Osb.)	Lab-made *Citrus* juice
Bitter orange (*Citrus aurantium* L.)	Lab-made *Citrus* juice
Mandarin (*Citrus deliciosa* Ten.)	Lab-made *Citrus* juice
Pink grapefruit (*Citrus paradise* McF.)	Lab-made *Citrus* juice
Commercial beverages	
Limoncello	Lemon peels 7%
Bergamino	Bergamot peels 8%
Cinnamon liquor	Cinnamon infusion 20%
Cocoa, sweet orange and cinnamon infusion	Sweet orange peel, bark Cinnamomum zeylanicum blume
*Citrus*-flavoured nonalcoholic beverage	Bergamot essential oil 0.2%, lemon infusion 0.2%
Bergamot-flavoured nonalcoholic beverage	Bergamot juice 13%
Green-mandarin-flavoured nonalcoholic beverage	Mandarin essential oil 1%, mandarin infusion 4%
Jams	
Lemon	Calabrian Lemon fruits
Bergamot	Bergamot pulp and peel
Orange	Orange pulp and peel
Orange and cinnamon	Organic orange 40%, powder cinnamon 0.1%
*Citrus* fruits and ginger	Organic *Citrus* fruits 40% (orange 48%, lemon 47%, mandarin 5%), organic ginger 0.9%
Bakery products	
Lemon-flavoured snack	Lemon flavour 1%
Lemon-flavoured biscuits	Lemon flavour 2.5%
Lemon- and almond-flavoured biscuits	Candied peel lemon 6.5%, lemon juice 0.4%
Mandarin- and almond-flavoured biscuits	Candied peel mandarin lemon 6.5%, mandarin juice 2%
Bergamot-flavoured biscuits	Bergamot 5%
Bergamot-flavoured dessert biscuits	Bergamot 5%

**Table 2 foods-12-03408-t002:** Amount of coumarins ^§^, furocoumarins * and polymethoxyflavones ^+^ (mg L^−1^ ± standard deviation) in the lab-made *Citrus* juices analysed.

Compound	Lemon	Lime	Bergamot	Citron	Sweet Orange	Bitter Orange	Mandarin	Pink Grapefruit
Isomeranzin (2) ^§^	<LoD	<LoD	<LoD	<LoD	<LoD	0.40 ± 0.012	<LoD	1.80 ± 0.062
Herniarin (3) ^§^	<LoD	3.24 ± 0.001	<LoD	<LoD	<LoD	<LoD	<LoD	<LoD
Oxypeucedanin (4) *	0.01 ± 0.000	0.58 ± 0.014	<LoD	0.06 ± 0.003	<LoD	<LoD	<LoD	<LoD
8-geranyloxypsoralen (5) *	0.38 ± 0.023	<LoQ	<LoD	<LoQ	<LoQ	<LoQ	<LoQ	<LoQ
Aurapten (7) ^§^	<LoD	<LoD	<LoD	<LoD	<LoD	<LoD	<LoD	0.26 ± 0.013
Meranzin (8) ^§^	<LoD	<LoD	<LoD	<LoD	<LoD	0.95 ± 0.008	<LoD	1.27 ± 0.009
Epoxyaurapten (10) ^§^	<LoD	<LoD	<LoD	<LoD	<LoD	<LoD	<LoD	0.25 ± 0.002
Phellopterin (12) *	<LoD	0.05 ± 0.002	<LoD	<LoD	<LoD	<LoD	<LoD	<LoD
Byakangelicol (13) *	0.02 ± 0.005	0.16 ± 0.005	<LoD	0.09 ± 0.005	<LoD	<LoD	<LoD	<LoD
Citropten (14) ^§^	0.07 ± 0.015	1.03 ± 0.020	<LoQ	0.02 ± 0.000	<LoQ	<LoQ	<LoQ	<LoQ
Cnidilin (15) *	<LoD	0.01 ± 0.000	<LoD	<LoD	<LoD	<LoD	<LoD	<LoD
Isopimpinellin (16) *	<LoD	1.65 ± 0.086	<LoD	<LoD	<LoD	<LoD	<LoD	<LoD
Bergapten (18) *	<LoD	0.50 ± 0.006	7.02 ± 0.062	<LoD	<LoD	0.37 ± 0.007	<LoD	0.01 ± 0.001
Isoimperatorin (19) *	<LoD	0.01 ± 0.000	<LoD	<LoD	<LoD	<LoD	<LoD	<LoD
Oxypeucedanin hydrate (20) *	0.09 ± 0.006	4.40 ± 0.121	<LoD	0.15 ± 0.007	<LoD	<LoD	<LoD	<LoD
5-geranyloxy-7-methoxycoumarin (21) ^§^	<LoQ	5.16 ± 0.018	0.05 ± 0.001	<LoQ	<LoQ	<LoQ	<LoD	<LoQ
Bergamottin (22) *	<LoQ	5.96 ± 0.338	15.81 ± 0.056	<LoQ	<LoQ	<LoQ	<LoQ	0.59 ± 0.036
Epoxybergamottin (23) *	<LoD	<LoD	<LoD	<LoD	<LoD	<LoD	<LoQ	0.84 ± 0.036
Byakangelicin (24) *	0.04 ± 0.001	5.37± 0.020	<LoD	0.20 ± 0.015	<LoD	<LoD	<LoQ	<LoD
Tangeretin (25) ^+^	<LoD	<LoD	<LoD	<LoD	0.05 ± 0.002	0.15 ± 0.002	7.31 ± 0.069	0.07 ± 0.005
Nobiletin (26) ^+^	<LoD	<LoD	0.01 ± 0.001	<LoD	0.78 ± 0.016	0.46 ± 0.019	18.79 ± 0.469	0.64 ± 0.028
Tetra-O-methylscutellarein (27) ^+^	<LoD	<LoD	0.01 ± 0.000	<LoD	<LoD	<LoD	0.40 ± 0.018	<LoD
Sinensetin (28) ^+^	<LoD	<LoD	<LoD	<LoD	<LoD	<LoD	84.10 ± 1.844	<LoD
TOT	1.26 ± 0.050	27.92 ± 0.631	22.90 ± 0.120	0.47 ± 0.030	0.83 ± 0.018	2.36 ± 0.48	110.54 ± 2.40	5.69 ± 0.192

Coumarin (1) ^§^, psoralen (6) *, 8-methoxypsoralen (9) *, cnidicin (11) * and meranzin hydrate (17) ^§^ had the lowest LoD scores among all the samples analysed.

**Table 3 foods-12-03408-t003:** Amount of coumarins ^§^, furocoumarins * and polymethoxyflavones ^+^ (mg L^−1^ ± standard deviation) in the commercial *Citrus* beverages analysed.

Compound	Alcoholic Beverages			Nonalcoholic Beverages			
	Limoncello	Bergamino	Cinnamon Liquor	Bergamot	Green Mandarin	*Citrus*	Infusion
Coumarin (1) ^§^	<LoD	<LoD	3.55 ± 0.075	<LoD	<LoD	<LoD	0.22 ± 0.006
Herniarin (3) ^§^	<LoD	0.10 ± 0.002	<LoD	0.05 ± 0.001	<LoD	<LoD	<LoD
Oxypeucedanin (4) *	0.33 ± 0.002	<LoD	<LoD	<LoD	<LoD	<LoD	<LoD
8-geranyloxypsoralen (5) *	7.39 ± 0.567	<LoQ	<LoQ	<LoD	<LoQ	<LoQ	<LoQ
Cnidicin (11) *	0.06 ± 0.003	<LoD	<LoD	<LoD	<LoD	<LoD	<LoD
Phellopterin (12) *	0.12 ± 0.004	<LoD	<LoD	<LoD	<LoD	<LoD	<LoD
Byakangelicol (13) *	0.80 ± 0.027	<LoD	<LoD	<LoD	<LoD	<LoD	<LoD
Citropten (14) ^§^	1.92 ± 0.067	6.00 ± 0.094	<LoQ	0.65 ± 0.031	<LoQ	0.01 ± 0.001	<LoQ
Cnidilin (15) *	0.04 ± 0.003	<LoD	<LoD	<LoD	<LoD	<LoD	<LoD
Isopimpinellin (16) *	<LoD	<LoD	<LoD	0.01 ± 0.002	<LoD	<LoD	<LoD
Bergapten (18) *	<LoD	3.08 ± 0.007	<LoD	3.09 ± 0.011	<LoD	<LoD	<LoD
Isoimperatorin (19) *	0.10 ± 0.003	<LoD	<LoD	<LoD	<LoD	<LoD	<LoD
Oxypeucedanin hydrate (20) *	4.00 ± 0.005	<LoD	<LoD	0.06 ± 0.001	<LoD	0.02 ± 0.001	<LoD
5-geranyloxy-7-methoxycoumarin (21) ^§^	2.40 ± 0.053	2.09 ± 0.115	<LoQ	0.06 ± 0.002	<LoQ	<LoQ	<LoQ
Bergamottin (22) *	3.22 ± 0.144	12.69 ± 0.059	<LoQ	0.20 ± 0.008	<LoQ	<LoQ	<LoQ
Byakangelicin (24) *	1.68 ± 0.089	<LoD	<LoD	<LoD	<LoD	<LoD	<LoQ
Tangeretin (25) ^+^	<LoD	<LoD	<LoD	<LoD	1.34 ± 0.054	<LoD	<LoQ
Nobiletin (26) ^+^	<LoD	0.07 ± 0.009	<LoD	0.12 ± 0.002	4.36 ± 0.132	<LoD	0.11 ± 0.002
Tetra-O-methylscutellarein (27) ^+^	<LoD	<LoD	<LoD	<LoD	<LoD	<LoD	0.04 ± 0.001
Sinensetin (28) ^+^	<LoD	<LoD	<LoD	<LoD	10.86 ± 0.599	<LoD	<LoQ
TOT	22.07 ± 0.967	24.03 ± 0.286	3.55 ± 0.075	4.72 ± 0.058	16.52 ± 0.785	0.03 ± 0.002	0.37 ± 0.009

Isomeranzin (2) ^§^, psoralen (6) *, aurapten (7) ^§^, meranzin (8) ^§^, 8-methoxypsoralen (9) *, epoxyaurapten (10) ^§^, meranzin hydrate (17) ^§^ and epoxybergamottin (23) * had the lowest LoD scores among all the samples analysed.

**Table 4 foods-12-03408-t004:** Amount of coumarins ^§^, furocoumarins * and polymethoxyflavones ^+^ (mg Kg^−1^ ± standard deviation) in the *Citrus* jams and bakery products analysed. For compound identification, see Table 2 and Table 3.

Compound	Jams					Bakery Products					
	Lemon	Bergamot	Orange	Orange Cinnamon	*Citrus* Ginger	Lemon Snack	Lemon Biscuits	Lemon Almond Biscuits	Mandarin Almond Biscuits	Bergamot Biscuits	Bergamot Dessert Biscuits
1 ^§^	<LoD	<LoD	<LoD	1.16 ± 0.073	<LoD	<LoD	<LoD	<LoD	<LoD	<LoD	<LoD
2 ^§^	<LoD	<LoD	<LoD	0.05 ± 0.002	<LoD	0.03 ± 0.002	<LoD	<LoD	<LoD	<LoD	<LoD
3 ^§^	<LoD	0.01 ± 0.001	<LoD	<LoD	<LoD	<LoD	<LoD	<LoD	<LoD	<LoD	<LoD
4 *	0.07 ± 0.003	<LoD	<LoD	<LoD	0.07 ± 0.002	<LoD	<LoD	<LoD	<LoD	<LoD	<LoD
5 *	6.78 ± 0.128	<LoQ	<LoQ	<LoQ	<LoQ	0.47 ± 0.021	0.45 ± 0.067	0.34 ± 0.038	<LoD	<LoQ	<LoQ
8 ^§^	<LoD	<LoD	<LoD	0.95 ± 0.008	<LoD	0.04 ± 0.001	<LoD	<LoD	<LoD	<LoD	<LoD
13 *	0.14 ± 0.012	<LoD	<LoD	<LoD	0.06 ± 0.001	<LoD	<LoD	<LoD	<LoD	<LoD	<LoD
14 ^§^	5.40 ± 2.814	3.92 ± 0.045	<LoQ	0.02 ± 0.000	0.22 ± 0.002	0.01 ± 0.000	0.04 ± 0.002	0.09 ± 0.002	<LoD	2.59 ± 0.079	0.03 ± 0.001
18 *	<LoD	<LoD	<LoD	0.04 ± 0.001	<LoD	<LoD	<LoD	<LoD	<LoD	2.29 ± 0.091	0.03 ± 0.000
19 *	<LoD	<LoD	<LoD	<LoD	<LoD	<LoD	<LoD	0.01 ± 0.000	<LoD	<LoD	<LoD
20 *	3.61 ± 0.004	<LoD	<LoD	<LoD	0.20 ± 0.010	0.01 ± 0.000	0.01 ± 0.000	0.02 ± 0.001	<LoD	<LoD	<LoD
21 ^§^	1.07 ± 0.001	1.08 ± 0.012	<LoQ	<LoQ	0.09 ± 0.001	0.02 ± 0.001	0.09 ± 0.003	0.37 ± 0.003	<LoQ	2.30 ± 0.091	0.04 ± 0.005
22 *	1.03 ± 0.000	10.96 ± 0.100	<LoQ	<LoQ	0.08 ± 0.000	0.04 ± 0.000	0.07 ± 0.006	0.34 ± 0.001	<LoD	11.18 ± 0.027	0.55 ± 0.008
24 *	2.31 ± 0.001	<LoD	<LoD	<LoD	0.14 ± 0.005	<LoD	<LoD	<LoD	<LoD	<LoD	<LoD
25 ^+^	<LoD	<LoD	<LoD	<LoD	<LoD	<LoD	<LoD	<LoD	1.00 ± 0.009	<LoD	<LoD
26 ^+^	<LoD	<LoD	2.00 ± 0.018	6.95 ± 0.036	4.59 ± 0.270	<LoD	<LoD	<LoD	1.26 ± 0.028	<LoD	<LoD
27 ^+^	<LoD	0.07 ± 0.006	0.76 ± 0.019	2.62 ± 0.032	1.63 ± 0.040	<LoD	<LoD	<LoD	0.28 ± 0.007	0.04 ± 0.001	<LoD
TOT	20.41 ± 2.963	16.02 ± 0.164	2.76 ± 0.037	12.19 ± 0.152	7.38 ± 0.330	0.62 ± 0.025	0.66 ± 0.078	1.26 ± 0.045	4.50 ± 0.044	18.38 ± 0.198	0.63 ± 0.014

Psoralen (6) *, aurapten (7) ^§^, 8-methoxypsoralen (9) *, epoxyaurapten (10) ^§^, cnidicin (11) *, phellopterin (12) *, cnidilin (15) *, isopimpinellin (16) *, meranzin hydrate (17) ^§^, epoxybergamottin (23) * and sinensetin (28) ^+^ had the lowest LoD scores among all the samples analysed.

## Data Availability

Not applicable.

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
