# Peer review of "Determination of Oxygen Heterocyclic Compounds in Foods Using Supercritical Fluid Chromatography–Tandem Mass Spectrometry"

_foods, 2023, doi:10.3390/foods12183408_

Round 1

Reviewer 1 Report

This paper demonstrates the first reported (to the authors’ knowledge) use of SFC-QqQ/MS for the determination of oxygen heterocyclic compounds in foods. Presented work seems to be a continuation of the authors' previous publications, as it includes previously developed and optimized SFC-MS and extraction methods. Thus, this paper focuses on analysing the different foods and interpretation of the data obtained.

I have a few comments requiring your attention.

1. I've found a several misprints:

line 14 - The aim of this research article was to oxygen - probably "was to determinate oxygen"

line 55 - the aim of traking the content

lines 82, 85 - coumarins furocoumarins and polymethoxyflavones - missing comma

line 173 - HPLC analysis of acqueos samples

line 181 - chromatograms of sected samples analysed

line 199 - are the most hydrofilc

line 209 - rapresent an optimum

line 292 - in twentyseven foods - it should be twenty-six, as stated in abstract

line 355 - pro le of coumarins, psoralens, and polymethoxy avones - profile, flavones

Table 1 - bakery - Mandarin and almond flavoured biscuits
Candied mandarin lemon 6.5% - probably "candied mandarin peel"

2. Line 88. This sentence seems unclear to me and should be rephraised. "Low consumption of organic solvent" can't be the mobile phase.

3. Tables 2,3,4. In my opinion, the value of a measurement result should end with the digit of the same place-value as its standard deviation value. For example, 0.38 ± 0.02.

4. Lines 179-180. This sentence also seems unclear. If it was possible to quantify analytes below the stated LoQs values (0.004-0.01 mg/L ), therefore, these limits are incorrect, and should be reduced. Or is it refers to "<LOQ" compounds?

5. I couldn't find reference number [30] in the manuscript.

Reviewer 2 Report

This study aims to determine oxygen heterocyclic compounds in twenty-six foods using supercritical fluid chromatography in combination with triple quadrupole mass spectrometry technique. The research is interesting and the results maybe help to exploit a fast and efficient method for quantitative analysis of coumarins, furocoumarins and polymethoxyflavones in foods.

However, major revision is recommended, and some other questions and suggestions are as following.

1 the abstract should be reorganized, this section should contain not only about the research significance but also the research results and conclusion.

2 when abbreviation was used for the first time, the full name should be given, please check the whole manuscript.

3 methodology validation was not well described in section 2, only simple words were used in line 163-164. How to carry out these experiments? And what are the results for LoDs, LoQs, accuracy, intra- and inter-day repeatability for your developed method?

4 Whether the experiment was repeated? how to carry out the statistical analysis?

5 section 3, this section should be results and discussion, and some references should be supplemented in introduction and discussion. please refer to the recently released publication https://doi.org/10.1016/j.fochx.2023.100628 in line 74-75

6 please check the spelling mistakes in your manuscript, for example, line 15 chromatography.  

7 please delete the general or unimportant words in the key words.

Author Response

Please see attachements

Round 2

Reviewer 2 Report

The authors have revised the manuscript with great efficiency. However, the most important comments are not addressed well, and need to be treated strictly and  patiently, such as question 1, 3, 4, 5. 

  •  

Author Response

Please see the attachment of the author's response.
